



# Ensemble Riemannian Data Assimilation: Towards High-dimensional Implementation

Sagar K. Tamang[1], Ardeshir Ebtehaj[1], Peter J. van Leeuwen[2], Gilad Lerman[3], and
Efi Foufoula-Georgiou[4]

[1]Department of Civil, Environmental and Geo-Engineering and Saint Anthony Falls Laboratory, University of
Minnesota-Twin Cities, Twin Cities, Minnesota, USA
[2]Department of Atmospheric Science, Colorado State University, Fort Collins, Colorado, USA
[3]School of Mathematics, University of Minnesota-Twin Cities, Twin Cities, Minnesota, USA
[4]Department of Civil and Environmental Engineering and Department of Earth System Science, University of California
Irvine, Irvine, California, USA

**Correspondence:** Sagar K. Tamang (taman011@umn.edu)

**Abstract.** This paper presents the results of the Ensemble Riemannian Data Assimilation for relatively high-dimensional nonlinear dynamical systems, focusing on the chaotic Lorenz-96 model and a two-layer quasi-geostrophic (QG) model of atmospheric circulation. The analysis state in this approach is inferred from a joint distribution that optimally couples the background probability distribution and the likelihood function, enabling formal treatment of systematic biases without any

Gaussian assumptions. Despite the risk of the curse of dimensionality in the computation of the coupling distribution, comparisons with the classic implementation of the particle filter and the stochastic ensemble Kalman filter demonstrate that with the same ensemble size, the presented methodology could improve the predictability of dynamical systems. In particular, under systematic errors, the root mean squared error of the analysis state can be reduced by 20% (30%) in Lorenz-96 (QG) model.

## 1 Introduction

The science of data assimilation (DA) aims to optimally combine the information content of observations with forecasts of Earth system models (ESM) to improve the estimation of their initial conditions and thus their predictive capabilities (Kalnay, 2003; Carrassi et al., 2018). Current DA methodologies, either variational (Lorenc, 1986; Zupanski, 1993; Courtier et al., 1994; Rabier et al., 2000; Poterjoy and Zhang, 2014) or filtering (Kalman, 1960; Bishop et al., 2001; Anderson, 2001; Tippett et al., 2003; Janjić et al., 2011; Carrassi and Vannitsem, 2011; Anderson and Lei, 2013; Lei et al., 2018), largely rely on penalization

of second-order statistics of the unbiased model and observation errors over the Euclidean space. For example, in the three-dimensional variational (3D-Var) DA (Lorenc, 1986; Courtier et al., 1998; Lorenc et al., 2000; Li et al., 2013), a least-squares cost function comprising of weighted Euclidean distances of the state from the previous model forecasts (background state) and the observations is formulated. Solution of this cost function leads to an analysis state, which is a weighted average of the forecasts and observations across multiple dimensions of the problem with the weights dictated by prescribed background and

observation error covariance matrices. The variants of the Kalman filtering DA methods (Evensen, 1994a; Reichle et al., 2002;





Evensen, 2003; Nerger et al., 2012b; Houtekamer and Zhang, 2016) also follow the same principle but in these methods, the background covariance contains information from past observations and model evolution.

Apart from the Euclidean distance, other measures and distance metrics including the quadratic mutual information (Kapur, 1994), Kullback-Leibler (KL) divergence (Kullback and Leibler, 1951), Hellinger distance (Hellinger, 1909), and Wasserstein
distance (Villani, 2003) have been also utilized in DA frameworks. Among others, Tagade and Ravela (2014) introduced a nonlinear filter, where the analysis is obtained through maximization of the quadratic mutual information. Maclean et al. (2017) utilized the Hellinger distance to measure the difference between the predicted and observed spatial patterns in oceanic flows. Chianese et al. (2018) introduced a variational DA method in which minimization of the KL divergence led to an approximation of the bias terms and model parameters. Similarly, Li et al. (2019) employed the KL divergence in an optimization framework to
incorporate inequality constraints in the Ensemble Kalman Filter (EnKF, Evensen, 1994b). Recently, Pulido and van Leeuwen (2019) developed a mapping particle filter in which particles are pushed towards the posterior density by minimizing the KL divergence between the posterior and a series of intermediate probability densities.

In filtering class of DA methodologies, coupling techniques have been proposed as an alternative to the classic Bayesian inference (El Moselhy and Marzouk, 2012; Spantini et al., 2019). El Moselhy and Marzouk (2012) presented a new approach
to find an optimal transport map that pushes forward the background to the posterior distribution. The approach was extended for generalization of the EnKF by deriving nonlinear coupling between the forecast and posterior distributions (Spantini et al., 2019). In recent years, the Wasserstein or the Earth mover's distance, originating from the theory of optimal mass transport (OMT, Monge, 1781; Kantorovich, 1942; Villani, 2003; Kolouri et al., 2017; Chen et al., 2017, 2018b, 2019a), has been also gaining attention in the DA community. Reich (2013) introduced a new resampling approach in particle filters, using the
OMT, to maximize the correlation between the prior and posterior ensemble members. Ning et al. (2014) further utilized the Wasserstein distance to treat position errors arising from uncertain model parameters. Following on this work, Feyeux et al. (2018) proposed to replace the weighted Euclidean distance with the Wasserstein distance in variational DA frameworks to treat position error. Tamang et al. (2020) proposed to use the Wasserstein distance to regularize a variational DA framework for treating systematic errors arising from the model forecast in chaotic systems.

However, DA frameworks utilizing the Wasserstein distance are computationally expensive as they require obtaining a joint distribution that couples two marginal distributions. Finding this joint distribution often relies on interior-point optimization methods (Altman and Gondzio, 1999) or the Orlin's algorithm (Orlin, 1993) that have super-cubic run time – making the Wasserstein DA computationally challenging even for relatively low dimensional problems. More recently, to reduce the computational cost, Tamang et al. (2021) used entropic regularization of the OMT formulation (Cuturi, 2013) through a new frame-
work, called Ensemble Riemannian Data Assimilation (EnRDA) to cope with systematic errors and tested it on a 3-dimensional Lorenz-63 model (Lorenz, 1963).

Unlike Euclidean DA with a known connection with the family of Gaussian distributions through Bayes' theorem, the EnRDA does not rely on any parametric assumptions about the input probability distributions. Therefore, it does not guarantee an analysis state with a minimum mean squared error. However, it enables to optimally (i) interpolate between the forecast





distribution and the normalized likelihood function without any parametric assumptions about their shapes and (ii) formally penalize systematic translations between them arising due to potential geophysical biases.

However, the computational complexity of finding an optimal joint coupling between two $m-$dimensional probability distributions supported on $d$ points in each dimension using the entropic regularization is $O(d^{2m})$. This might impose a significant limitation on the direct use of EnRDA for high-dimensional geophysical problems, where the problem dimension easily ex-

ceed millions. As will be discussed later, the joint distribution in EnRDA is sampled at $N^2$ support points, with $N$ number of ensembles, reducing the computational complexity to $O(N^2)$ at the expense of losing accuracy in optimal estimation of the joint distribution. Therefore, beyond implementation on a low-dimensional dynamical system, such as the 3-dimensional Lorenz-63, the key questions that we aim to answer are as follows: Does the effectiveness of the EnRDA implementation still remain valid on high-dimensional DA problems where the ensemble size is smaller than the problem dimension? How does

EnRDA perform, under systematic errors, in comparison to classic ensemble DA techniques with comparable ensemble size? To answer the above questions, we implement EnRDA on the relatively high-dimensional chaotic Lorenz-96 system (Lorenz, 1995) and a two-layer quasi-geostrophic (QG) model of atmospheric circulation (Pedlosky et al., 1987). The results demonstrate that EnRDA can potentially enhance predictability of high-dimensional geophysical systems, when the state variables are not necessarily Gaussian and are corrupted with systematic errors.

The outline of the paper is as follows. Section 2 provides a brief background on optimal mass transport and Wasserstein distance. A brief review of the EnRDA methodology is presented in Section 3. Section 4 presents different test cases of implementation on the Lorenz-96 and the QG model and documents the performance of the presented approach in comparison with the classic implementation of the standard particle filter with resampling and the Stochastic Ensemble Kalman Filter (SEnKF). A summary and concluding remarks are presented in Section 5. The details of the entropic regularization for the

EnRDA, and covariance inflation and localization procedures for the SEnKF are provided in Appendix A.

## 2   Background on OMT and the Wasserstein Barycenter

We provide a brief background on the theory of optimal mass transport (OMT) and Wasserstein barycenters. The OMT theory, first put forward by Monge (1781), aims to find the minimum cost of transporting distributed masses of materials from known source points to target points. The theory was later expanded as a new tool to compare probability distributions (Brenier, 1987;

Villani, 2003) and since then has found its applications in the field of data assimilation (Ning et al., 2014; Feyeux et al., 2018; Li et al., 2018; Tamang et al., 2020), subsurface geophysical inverse problems (Chen et al., 2018a; Yang et al., 2018; Yang and Engquist, 2018; Yong et al., 2019) and comparisons of climate model simulations (Vissio et al., 2020).

Let us consider a discrete source probability distribution $p(\mathbf{x}) = \sum_{i=1}^{M} p_{\mathbf{x}_i} \delta_{\mathbf{x}_i}$ and a target distribution $p(\mathbf{y}) = \sum_{j=1}^{N} p_{\mathbf{y}_j} \delta_{\mathbf{y}_j}$ with their respective probability masses $\{\mathbf{p}_x \in \mathbb{R}_+^M : \sum_i p_{\mathbf{x}_i} = 1\}$ and $\{\mathbf{p}_y \in \mathbb{R}_+^N : \sum_j p_{\mathbf{y}_j} = 1\}$ supported on $m$- and $n$-

element column vectors $\mathbf{x}_i \in \mathbb{R}^m$ and $\mathbf{y}_j \in \mathbb{R}^n$, respectively. The notation $\mathbf{p}_x \in \mathbb{R}_+^M$ represents probability masses $\mathbf{p}_x$ containing non-negative real numbers supported on $M$ points, whereas $\delta_{\mathbf{x}}$ is the Dirac function at $\mathbf{x}$. In the Monge formulation, the goal is to seek an optimal surjective transportation map $T_{\#}^a p(\mathbf{x}) = p(\mathbf{y})$ that "pushes forward" the source distribution $p(\mathbf{x})$





towards the target distribution $p(\mathbf{y})$, with a minimum transportation cost as follows:

$$T^a = \underset{T}{\operatorname{argmin}} \sum_{i=1}^{M} c(\mathbf{x}_i, T(\mathbf{x}_i)) \quad \text{s.t. } T^a_\# p(\mathbf{x}) = p(\mathbf{y}), \tag{1}$$

where $c(\cdot, \cdot) \in \mathbb{R}_+$ represents the cost of transporting a unit mass from one support point in $\mathbf{x}$ to another one in $\mathbf{y}$.

The problem formulation by Monge as expressed in Equation 1, however, is non-convex and the existence of an optimal transportation map is not guaranteed (Chen et al., 2019b) – especially, when the number of support points for the target distribution exceeds that of the source distribution ($N > M$) (Peyré et al., 2019). This limitation was overcome by Kantorovich (1942) who introduced a probabilistic formulation of OMT – allowing splitting of probability mass from a single source point to

multiple target points. The Kantorovich formalism recasts the OMT problem in a linear programming framework that finds an optimal joint distribution or coupling $\mathbf{U}^a \in \mathbb{R}_+^{M \times N}$ that couples the marginal source and target distributions with the following optimality criterion:

$$\mathbf{U}^a = \underset{\mathbf{U}}{\operatorname{argmin}} \operatorname{tr}(\mathbf{C}^{\mathrm{T}} \mathbf{U}) \quad \text{s.t.} \quad \begin{cases} \mathbf{U} \in \mathbb{R}_+^{M \times N} \\ \mathbf{U} \mathbb{1}_N = \mathbf{p}_x \\ \mathbf{U}^{\mathrm{T}} \mathbb{1}_M = \mathbf{p}_y \end{cases}, \tag{2}$$

where $\operatorname{tr}(\cdot)$ is the trace of a matrix, $(\cdot)^{\mathrm{T}}$ is the transposition operator and $\mathbb{1}_M$ represents an $M$-element column vector of ones. In

the above formulation, the known $\{\mathbf{C} \in \mathbb{R}_+^{M \times N} : c_{ij} = \|\mathbf{x}_i - \mathbf{y}_j\|_2^2\}$ denotes the so-called transportation cost matrix which is defined based on the $\ell_2$-norm $\|\cdot\|_2$ or the Euclidean distance between the support points of the source and target distributions. Here, the $(i,j)^{\text{th}}$ element $u_{ij}^a$ of optimal joint distribution $\mathbf{U}^a$ represents the respective amount of mass transported from support point $\mathbf{x}_i$ to $\mathbf{y}_j$. Then, the 2-Wasserstein distance or metric between the marginal probability distributions is defined as the square root of the optimal transportation cost $d_{\mathcal{W}}(\mathbf{p}_x, \mathbf{p}_y) = \left(\operatorname{tr}(\mathbf{C}^{\mathrm{T}} \mathbf{U}^a)\right)^{\frac{1}{2}}$ (Dobrushin, 1970; Villani, 2008). It should be

noted that due to the linear equality and non-negativity constraints in Equation 2, the family of joint distributions that satisfy these constraints forms a bounded convex polytope (Cuturi and Peyré, 2018) and consequently, the optimal joint distribution $\mathbf{U}^a$ is located on one of the extreme points of such a polytope (Peyré et al., 2019).

Recalling that over the Euclidean space, the barycenter of a group of points is equivalent to their (weighted) mean value. The Wasserstein metric offers a Riemannian generalization of this problem and allows to define the barycenter of a family

of probability distributions (Rabin et al., 2011; Bigot et al., 2012; Srivastava et al., 2018). In particular, for a group of $K$ probability mass functions $\mathbf{p}_1, \ldots, \mathbf{p}_K$, a Wasserstein barycenter $\mathbf{p}_\eta$ is defined as their Fréchet mean (Fréchet, 1948) as follows (Agueh and Carlier, 2011):

$$\mathbf{p}_\eta = \underset{\mathbf{p}}{\operatorname{argmin}} \sum_{k=1}^{K} \eta_k d_{\mathcal{W}}^2(\mathbf{p}, \mathbf{p}_k), \tag{3}$$

where $\{(\eta_1, \ldots, \eta_K)^{\mathrm{T}} \in \mathbb{R}_+^K : \sum_k \eta_k = 1\}$ represent the weights associated with the respective distributions. In special cases

where the group of $K$ distributions is Gaussian $\{\mathcal{N}(\boldsymbol{\mu}_1, \boldsymbol{\Sigma}_1), \ldots, \mathcal{N}(\boldsymbol{\mu}_K, \boldsymbol{\Sigma}_K)\}$ with mean $\boldsymbol{\mu}_1, \ldots, \boldsymbol{\mu}_K$ and positive definite





covariance $\boldsymbol{\Sigma}_1, \ldots, \boldsymbol{\Sigma}_K$, the Wasserstein barycenter is also a Gaussian density $\mathcal{N}(\boldsymbol{\mu}_\eta, \boldsymbol{\Sigma}_\eta)$ with $\boldsymbol{\mu}_\eta = \sum_k \eta_k \boldsymbol{\mu}_k$ and $\boldsymbol{\Sigma}_\eta$ is the

unique positive definite root of the matrix equation $\boldsymbol{\Sigma} = \sum_k \eta_k \big(\boldsymbol{\Sigma}^{\frac{1}{2}} \boldsymbol{\Sigma}_k \boldsymbol{\Sigma}^{\frac{1}{2}}\big)^{\frac{1}{2}}$ (Agueh and Carlier, 2011).

## 3 Ensemble Riemannian Data Assimilation (EnRDA)

In this section, to be self content, we provide a brief summary of the EnRDA methodology while more details can be found
in (Tamang et al., 2021). Let us assume that the evolution of the $i^{\text{th}}$ ensemble member $\mathbf{x}_i \in \mathbb{R}^m$ of ESM simulations can be
presented as the following stochastic dynamical system:

$$\mathbf{x}_i^t = \mathcal{M}(\mathbf{x}_i^{t-1}) + \boldsymbol{\omega}_i^t \qquad i = 1, \ldots, M \,, \tag{4}$$

where $\mathcal{M} : \mathbb{R}^m \to \mathbb{R}^m$ is the deterministic nonlinear model operator, evolving the model state in time with a stochastic error
term $\boldsymbol{\omega}_i^t \in \mathbb{R}^m$. This dynamical system is observed at time $t$ through an observation equation $\mathbf{y}^t = \mathcal{H}(\mathbf{x}^t) + \boldsymbol{\upsilon}^t$, where $\mathcal{H} :$
$\mathbb{R}^m \to \mathbb{R}^n$ maps the state to the observation space and $\boldsymbol{\upsilon}^t \in \mathbb{R}^n$ represents an additive observation error. Note that the error
terms are not necessarily drawn from Gaussian distributions but need to have finite second-order moments.

Hereafter, we drop the time superscript for brevity and represent the model (or background) probability distribution as $p(\mathbf{x}) = \sum_{i=1}^M p_{\mathbf{x}_i} \delta_{\mathbf{x}_i}$ with its probability mass vector $\{\mathbf{p}_x \in \mathbb{R}_+^M : \sum_i p_{\mathbf{x}_i} = 1\}$. Furthermore, the normalized likelihood function is
represented as $\widetilde{p}(\mathbf{y}|\mathbf{x})$ centered at the given observation $\mathbf{y}$ with its probability mass vector $\{\widetilde{\mathbf{p}}_{y|x} \in \mathbb{R}_+^N : \sum_j \widetilde{p}_{\mathbf{y}|\mathbf{x}_j} = 1\}$. The
probability distribution of the analysis state $p(\mathbf{x}_a)$, is then defined as the Wasserstein barycenter between forecast distribution
and the normalized likelihood function:

$$p(\mathbf{x}_a) = \underset{p(\mathbf{z})}{\operatorname{argmin}} \left\{ \eta \, d_{\mathcal{W}}^2[p(\mathbf{x}), p(\mathbf{z})] + (1 - \eta) \, d_{\mathcal{W}}^2[\widetilde{p}(\mathbf{y}|\mathbf{x}), p(\mathbf{z})] \right\} \,, \tag{5}$$

where $\eta \in [0, 1]$ is a displacement parameter that controls the relative weight of the background and observation. The displace-
ment parameter $\eta$ is a hyperparameter that captures the relative weights of the histogram of the background state and likelihood
function in characterization of the analysis state distribution as a Wasserstein barycenter. The optimal value of $\eta$ needs to be
determined offline, using reference data through cross-validation studies. It is important to note that the above formalism re-
quires all dimensions to be observable and thus those dimensions with no observations cannot be updated, which is a limitation
of the current formalism compared to the Euclidean DA. This limitation is further discussed later on in Section 5.

To solve the above DA problem, we need to characterize the background distribution and the normalized likelihood function.
Similar to the approach used in particle filter (Gordon et al., 1993; van Leeuwen, 2010), we suggest approximating them through
ensemble realizations. For constructing the histogram of the normalized likelihood function, we can draw $N$ samples at each
assimilation cycle by perturbing the available observation $\mathbf{y}$ with the observation error $\mathcal{N}(0, \mathbf{R})$.





To obtain the Wasserstein barycenter $p(\mathbf{x}_a)$ in Equation 5, we use the McCann's formalism (McCann, 1997; Peyré et al., 2019):

$$p(\mathbf{x}_a) = \sum_{i=1}^{M} \sum_{j=1}^{N} u_{ij}^a \, \delta_{\mathbf{z}_{ij}}, \tag{6}$$

where $\mathbf{z}_{ij} = \eta \mathbf{x}_i + (1-\eta) \mathbf{y}_j$ represents the support points of the analysis distribution and $u_{ij}^a$ are the elements of the joint distribution $\{\mathbf{U}^a \in \mathbb{R}_+^{M \times N} : \sum_i \sum_j u_{ij} = 1\}$. It is important to note that the analysis state histogram, at each assimilation cycle, is supported on at most $M + N - 1$ points, which is the maximum number of non-zero entries in the optimal joint coupling (Peyré et al., 2019). To keep the number of ensemble members constant throughout, $M$ ensemble members are resampled from $p(\mathbf{x}_a)$ using the multinomial resampling scheme (Li et al., 2015).

Computation of the joint distribution in Equation 2 is computationally expensive as explained previously and can be prohibitive for high-dimensional geophysical problems. As suggested by Cuturi (2013), to reduce the computational cost, we regularize the cost function in the optimal transportation plan formulation of EnRDA by a Gibbs-Boltzmann entropy function:

$$\mathbf{U}^a = \underset{\mathbf{U}}{\mathrm{argmin}} \ \mathrm{tr}(\mathbf{C}^{\mathsf{T}} \mathbf{U}) - \gamma \, \mathrm{tr}\big(\mathbf{U}^{\mathsf{T}}[\log(\mathbf{U} - \mathbb{1}_M \mathbb{1}_N^{\mathsf{T}})]\big) \quad \text{s.t.} \quad \begin{cases} \mathbf{U} \in \mathbb{R}_+^{M \times N} \\ \mathbf{U} \mathbb{1}_N = \mathbf{p}_x \\ \mathbf{U}^{\mathsf{T}} \mathbb{1}_M = \widetilde{\mathbf{p}}_{y|x} \end{cases}, \tag{7}$$

where $\gamma \in \mathbb{R}_+$ is a regularization parameter. The entropic regularization transforms the original OMT formulation to a strictly convex problem, which can then be efficiently solved using Sinkhorn's algorithm (Sinkhorn, 1967). The details of Sinkhorn's algorithm for solving regularized optimal transportation problems are presented in Appendix A1. The regularization parameter $\gamma$ balances the solution between the optimal joint distribution and the one that maximizes the relative entropy. It is evident from Equation 7 that at the limit $\gamma \to 0$, the solution of Equation 7 converges to the analysis joint distribution with a minimum morphing cost. However, as the value of $\gamma$ increases, the convexity of the problem also increases, enabling the deployment of more efficient optimization algorithms than classic solvers of linear programming problems (Dantzig et al., 1955; Orlin, 1993). At the same time, the number of non-zero entries of the joint coupling increases from $M + N - 1$ to $MN$ points as $\gamma \to \infty$, which results in a maximum entropy solution that converges to $\mathbf{U}^a \to \mathbf{p}_x \widetilde{\mathbf{p}}_{y|x}^{\mathsf{T}}$. For a more comprehensive explanation of EnRDA, one can refer to Tamang et al. (2021).

As an example, we examine here the solution of Equation 5 between a banana-shaped distribution denoted by $\mathcal{F}(\xi_1, \xi_2, \xi_3, b) \propto \exp\big(-\xi_1(4 - bx_1 - x_2^2) - \xi_2(x_1^2 - \xi_3 x_2^2)\big)$ and a bivariate Gaussian distribution as a function of the displacement parameter $\eta \in [0,1]$ – resembling the background distribution $p(\mathbf{x})$ and the normalized likelihood function $\widetilde{p}(\mathbf{y}|\mathbf{x})$, respectively with regularization parameter $\gamma = 1000$. As seen from Figure 1, for lower values of $\eta$, the analysis state distribution is closer to the observation and its shape resembles the Gaussian distribution. However, as the value of $\eta$ increases, the analysis state distribution moves closer to the background distribution and starts morphing into a banana-shaped distribution. Therefore, the analysis state distribution is defined as the one that is sufficiently close to the background distribution and the


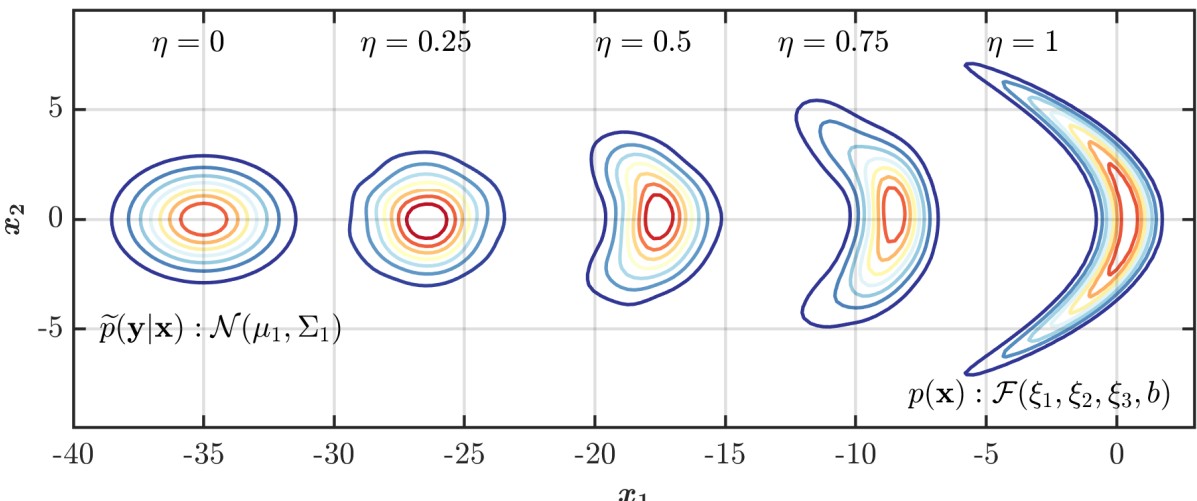

**Figure 1.** The analysis distribution obtained as a Wasserstein barycenter for different values of the displacement parameter $\eta \in [0, 1]$ between a background distribution represented by a banana-shaped distribution $p(\mathbf{x}) : \mathcal{F}(\xi_1, \xi_2, \xi_3, b)$ with $\xi_1 = 0.02$, $\xi_2 = 0.06$, $\xi_3 = 1.6$, and $b = 8$, and the normalized likelihood function represented by a bivariate Gaussian $\widetilde{p}(\mathbf{y}|\mathbf{x}) : \mathcal{N}(\boldsymbol{\mu}_1, \boldsymbol{\Sigma}_1)$, where $\boldsymbol{\mu}_1 = \begin{bmatrix} -35 \\ 0 \end{bmatrix}$ and $\boldsymbol{\Sigma}_1 = \begin{bmatrix} 3 & 0 \\ 0 & 2 \end{bmatrix}$.

normalized likelihood function not only based on their shape but also their central location – depending on the displacement parameter. Thus, unlike the Euclidean barycenter, this approach does not guarantee that the mean or mode of the analysis state probability distribution is a minimum mean-squared error estimate of the initial condition.

It is important to note that in the original OMT formulation, the number of support points required for the optimal joint coupling $\mathbf{U}$ scales with the problem dimension ($d^m$) making it potentially restrictive for the high-dimensional problems, where $d$ represents the number of support points in each dimension and $m$ is the number of dimensions. The presented EnRDA setting bypasses this problem by sampling the joint distribution using only $N$ ensemble members. However, such approximation might lead to errors in optimal estimation of the joint coupling that are translated into the analysis state. In the next section, we

present results from systems of well-known dynamics to investigate whether EnRDA can lead to a proper approximation of the analysis state under systematic errors in relatively high-dimensional nonlinear problems, when compared to classic ensemble DA approaches with comparable ensemble size.

## 4   Numerical Experiments and Results

### 4.1   Lorenz-96

The Lorenz model (Lorenz-96, Lorenz, 1995), which is widely adopted as a testbed for numerous DA experiments (Trevisan and Palatella, 2011; Tang et al., 2014; Shen and Tang, 2015; Lguensat et al., 2017; Tian et al., 2018), offers a simplified





representation of the extra-tropical dynamics in the Earth's atmosphere. The model coordinates $\{\mathbf{x} = (x_1, \ldots, x_K)^T \in \mathbb{R}^K\}$ at $K$ dimensions represent the state of an arbitrary atmospheric quantity measured along the Earth's latitudes at $K$ equally spaced longitudinal slices. The model is designed to mimic the continuous-time variation in atmospheric quantities due to

interactions between three major components namely advection, internal dissipation, and external forcing. The model dynamics is represented as follows:

$$\frac{dx_k}{dt} = (x_{k+1} - x_{k-2}) x_{k-1} - x_k + F, \qquad k = 1, \ldots, K, \tag{8}$$

where $F \in \mathbb{R}_+$ is a constant external forcing independent of the model state. The Lorenz-96 model has cyclic boundaries with $x_{-1} = x_{K-1}$, $x_0 = x_K$, and $x_{K+1} = x_1$. It is known that for small values of $F < 8/9$, the system approaches a steady state

condition with each coordinate value converging to the external forcing $x_k \to F$, $\forall k$, whereas for $F > 8/9$, chaos develops (Lorenz and Emanuel, 1998). For standard model setup with $F = 8$, the system is known to exhibit highly chaotic behavior with the largest Lyapunov exponent of 1.67 (Brajard et al., 2020).

### 4.1.1 Experimental Setup, Results and Discussion

We focus on the 40-dimensional Lorenz-96 system (i.e. $K = 40$) and compare EnRDA results with the classic implementation

of the particle filter (PF, Gordon et al., 1993; Van Leeuwen, 2009; van Leeuwen, 2010; Poterjoy and Anderson, 2016) and the Stochastic Ensemble Kalman filter (SEnKF, Evensen, 1994b; Houtekamer and Mitchell, 1998; Burgers et al., 1998; Janjić et al., 2011; Anderson, 2016; Van Leeuwen, 2020). Similar to the experimental setting suggested in (Lorenz and Emanuel, 1998; Nerger et al., 2012a), we initialize the model by choosing $x_{20} = 8.008$ and $x_k = 8$ for all other model coordinates. In order to avoid any initial transient effect, the model in Equation 8 is integrated for 1000 time steps using the fourth-order

Runge-Kutta approximation (Runge, 1895; Kutta, 1901) with a non-dimensional time step of $\Delta t = 0.01$ and the endpoint of the run is utilized as the initial condition for DA experimentation.

Similar to the suggested experimental setting in (van Leeuwen, 2010), we obtain the ground truth by integrating Equation 8 with a time step of $\Delta t$ over a time period of $T = 0$–$20$ in the absence of any model error. The observations are assumed to be available at each assimilation time interval of $10\Delta t$ and deviated from the ground truth by a Gaussian error $\boldsymbol{v}_t \sim \mathcal{N}(0, \sigma_{\text{obs}}^2 \boldsymbol{\Sigma}_\rho)$,

with $\sigma_{\text{obs}}^2 = 1$ and the correlation matrix $\boldsymbol{\Sigma}_\rho \in \mathbb{R}_+^{40 \times 40}$ with 1 on the diagonals, 0.5 on the first sub- and super-diagonals, and 0 everywhere else. The observation time step of $10\Delta t$ is equivalent to 12 hours in global ESMs (Lorenz, 1995).

To characterize the distribution of the background state for each DA methodology, 50 (5000) ensemble members (particles) for the SEnKF and EnRDA (PF) are generated using model errors $\boldsymbol{\omega}_t \sim \mathcal{N}(0, \sigma_t^2 \mathbf{I}_{40})$ with $\sigma_t^2 = 0.25$ for $t > 0$ and $\sigma_0^2 = 4$, where throughout $\mathbf{I}_m$ represents an $m \times m$ identity matrix. To alleviate the known degeneracy problem in the PF, a higher

number of particles was used. Furthermore, to introduce additional systematic background error, we utilize an erroneous external forcing of $F_m = 6$ instead of the "true" forcing value $F = 8$. To have a robust inference, the average values of the error metrics are reported for 50 experiments using different random realizations. As will be elaborated later on, we set the EnRDA displacement parameter $\eta = 0.44$, determined through a cross-validation study based on a minimum mean-squared error criterion. This tuning is similar to tuning inflation and localization parameters in a typical EnKF, or tuning length-scales


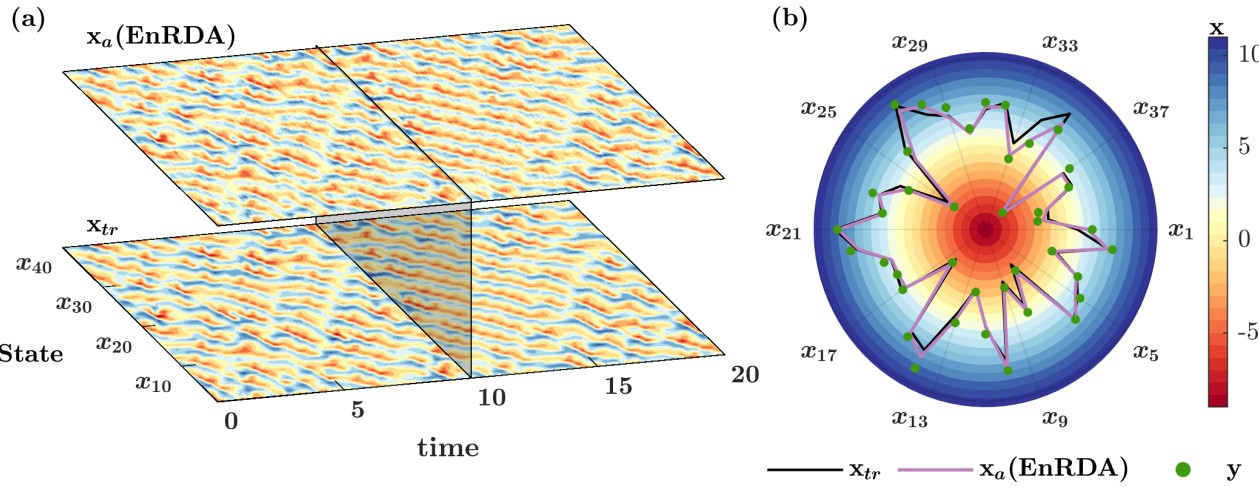

**Figure 2.** (a) Temporal evolution of the ground truth $\mathbf{x_{tr}}$ and analysis state $\mathbf{x}_a$ by ensemble Riemannian data assimilation (EnRDA) for $K = 40$ dimensions of the Lorenz-96 over $T = 0\text{--}20$ [t] and (b) their snapshots at $T = 10$ [t] together with the available observations $\mathbf{y}$.

in 3D- or 4D-Var. Note that we already introduced some systematic error because the truth has zero model error, while the prior does have model errors. In a fully unbiased set up the truth and the prior are drawn from the same distribution.

The results of EnRDA are shown in Figure 2. In the left panel, the temporal evolution of the ground truth and EnRDA analysis state is shown over all dimensions of the Lorenz-96, while a snapshot at time 10 [t] is presented in the right panel. The analysis state obtained from EnRDA follows the ground truth reasonably well during all time steps with a root mean squared error (rmse) of 0.85. The comparison of EnRDA with the classic implementations of the SEnKF and PF are shown in Figure 3 (a–c). It can be seen that the rmse of the PF increases sharply over time, suggesting that the problem of filter degeneracy still exists despite the higher number of particles. This problem is exacerbated due to the presence of bias causing a rapid collapse of the ensemble variance over time as more particles fall outside of the support set of the likelihood function. The root mean squared error of both the SEnKF and EnRDA is stabilized over time and is smaller by ∼20% (80%) in EnRDA compared to the SEnKF (PF). It is important to note that the presence of systematic bias due to erroneous choice of the external forcing inherently favors EnRDA over SEnKF since the latter is a minimum variance unbiased estimator at the limit $M \to \infty$, where $M$ represents the number of ensemble members.





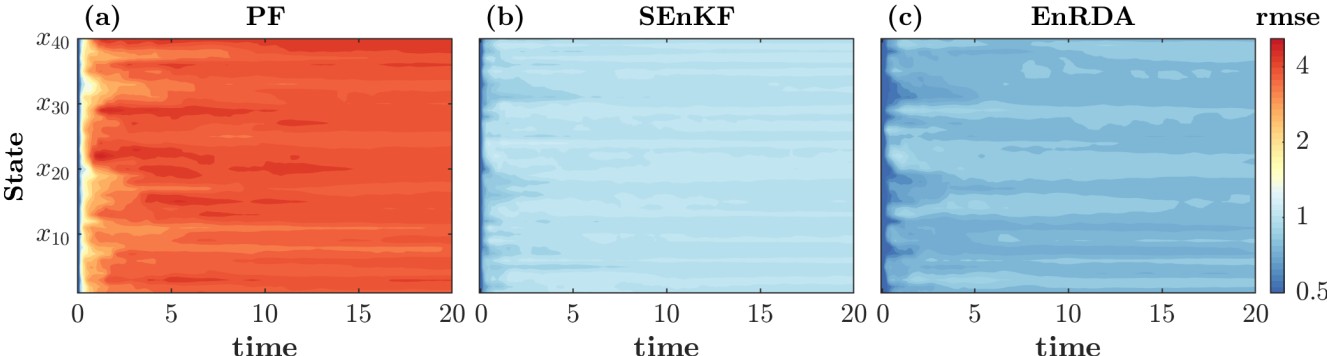

**Figure 3.** Temporal evolution of the root mean squared error (rmse) for the (a) Particle Filter (PF) with 5000 particles, (b) Stochastic Ensemble Kalman Filter (SEnKF), and (c) Ensemble Riemannian Data Assimilation (EnRDA) each with 50 ensemble members in 40-dimensional Lorenz-96 system. The results report the mean values of 50 independent simulations.

As previously noted, the displacement parameter $\eta$ plays an important role in EnRDA as it controls the shape and position of the analysis state distribution relative to the background distribution and the normalized likelihood function. Currently, there exists no known closed-form solution for optimal approximation of this parameter. Therefore, in this paper, we focus on determining its optimal value through heuristic cross-validation by an offline bias-variance trade-off analysis. Specifically, we quantify the rmse of the EnRDA analysis state for different values of $\eta$ for 50 independent simulations.

The bias and rmse, together with their respective $5^{\text{th}}$–$95^{\text{th}}$ percentile bounds, as functions of the displacement parameter $\eta$ are shown in Figure 4a. As explained earlier, when $\eta$ increases, the analysis distribution moves towards the background distribution. Since the background state is systematically biased due to the erroneous external forcing, the analysis bias increases monotonically with $\eta$; while the rmse shows a minimum point. Therefore, there exists a form of bias-variance trade-off in the analysis error, which leads to an approximation of an optimal value of $\eta$ based on a minimum rmse criterion. It is important to note that the background uncertainty and thus the optimal value of $\eta$ varies in response to the ensemble size as shown in Figure 4b. The reason is that a larger number of ensemble members reduces the uncertainty in the characterization of the background, but the bias is not affected. To compensate, a larger optimal value for $\eta$ is needed. This optimal value approaches an asymptotic value as the ensemble sample size increases and will achieve the highest value at the limit $M \rightarrow \infty$, when the sample moments converge to the biased forecast moments.

One may argue that such a tuning favors EnRDA since it explicitly accounts for the effects of bias, either in background or observations, while there is no bias correction mechanism in the implementation of the SEnKF and the PF. To make a fairer comparison, we investigate an alternative approach to approximate the displacement parameter solely based on the known error covariance matrices at each assimilation cycle. Recalling that in classic DA, the analysis state is essentially the Euclidean barycenter, where the relative weights of the background state and observations are optimally characterized based on the error covariances under zero bias assumptions. However, over the Wasserstein space, the displacement parameter determines the weight between the entire distribution of the background and the normalized likelihood function. Theoretically, knowing the

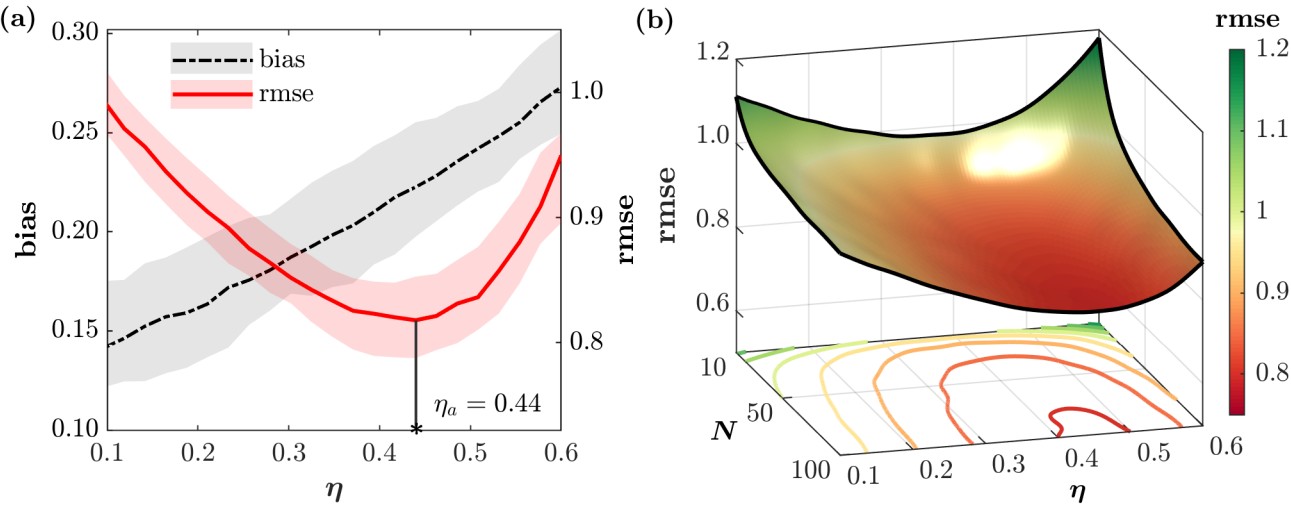

**Figure 4.** (a) Bias and root mean squared error (rmse) for a range of displacement parameter $\eta \in [0.1, 0.6]$ in Ensemble Riemannian Data Assimilation (EnRDA) with 50 ensemble members, obtained across 40-dimensions of the Lorenz-96 system. The shaded regions indicate the $5^{\text{th}}$–$95^{\text{th}}$ percentile bound for the respective error metrics obtained from 50 independent simulations. (b) Variation of rmse as a function of the number of ensemble members and $\eta$.

Wasserstein distances from ground truth to both likelihood function and forecast distribution enables to obtain an optimal value for $\eta$. Even though such distances are not known in reality, the total Wasserstein distance between the normalized likelihood function and the forecast distribution is known at each assimilation cycle. Therefore, given an estimate of the distance between the ground truth and the normalized likelihood function or the forecast distribution, leads to an approximation of $\eta$.

It is known that the square of the Wasserstein distance between two equal-mean Gaussian distributions $\mathcal{N}(\boldsymbol{\mu}, \boldsymbol{\Sigma}_1)$ and
$\mathcal{N}(\boldsymbol{\mu}, \boldsymbol{\Sigma}_2)$ is $d_{\mathcal{W}}^2 = \text{tr}(\boldsymbol{\Sigma}_1 + \boldsymbol{\Sigma}_2 - 2(\boldsymbol{\Sigma}_1^{\frac{1}{2}}\boldsymbol{\Sigma}_2\boldsymbol{\Sigma}_1^{\frac{1}{2}})^{\frac{1}{2}})$ (Chen et al., 2019b). Therefore, under the assumption that only the background state is biased, the square of the Wasserstein distance between the true state $\mathbf{x}_{tr}$, as a Dirac delta function, and the normalized likelihood function reduces to $\text{tr}(\mathbf{R})$. At the same time, the square of the Wasserstein distance between the normalized likelihood function and forecast distribution is $\text{tr}(\mathbf{C}^{\text{T}}\mathbf{U}^a)$. Therefore, we can approximate the interpolation parameter as $\overline{\eta_a} = \text{tr}(\mathbf{R})\left(\text{tr}(\mathbf{C}^{\text{T}}\mathbf{U}^a) + \text{tr}(\mathbf{R})\right)^{-1}$ without any explicit *a priori* knowledge of bias.

Comparisons of the rmse values for the studied DA methodologies as a function of ensemble size are shown in Figure 5. For EnRDA, the displacement parameter is obtained from the bias-aware cross-validation ($\eta = 0.44$, EnRDA-I) and from the known error covariances as explained above (EnRDA-II). The SEnKF and EnRDA result in smaller error metrics with a much smaller ensemble size than PF. As seen, EnRDA can perform well even for smaller ensemble sizes as low as 20. Its results quickly stabilize with more than 40 ensemble members and exhibit a marginal improvement over the SEnKF (12–24%) in the
presence of bias. The rmse of the SEnKF also stabilizes quickly but remains above the standard deviation of the observation error indicating that in the presence of bias, the lowest possible variance, known as the Cramer-Rao Lower Bound (Cramér, 1999; Rao et al., 1973) cannot be met.


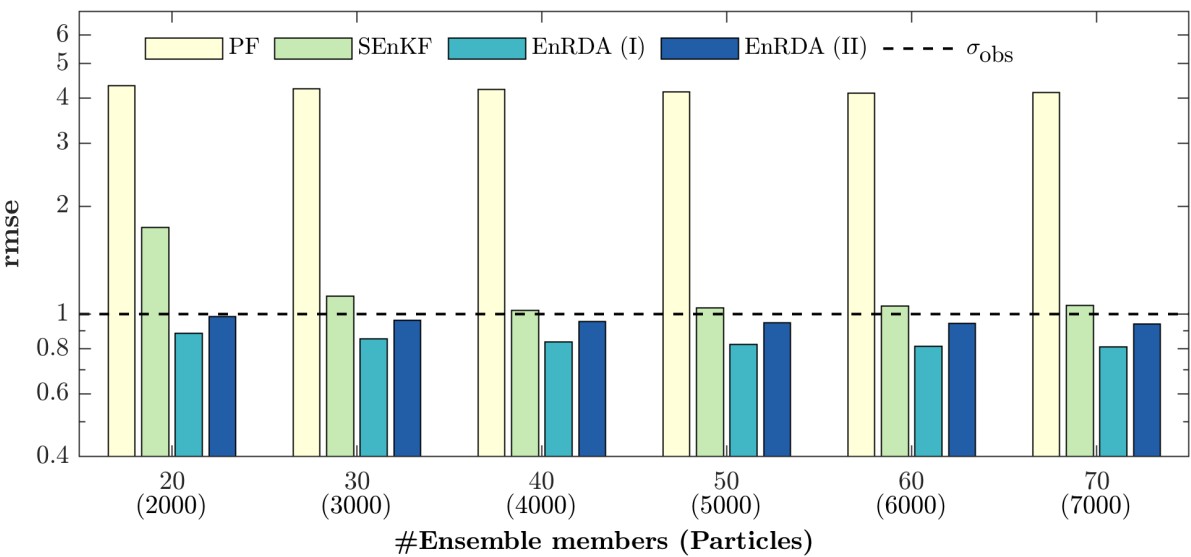

**Figure 5.** The root mean squared error (rmse) for the different number of ensemble members/particles in the Particle Filter (PF), Stochastic Ensemble Kalman Filter (SEnKF), and Ensemble Riemannian Data Assimilation (EnRDA) when the displacement parameter is obtained from bias-aware cross-validation (ENRDA-I) and a dynamic approach without *a priori* knowledge of bias (EnRDA-II) for Lorenz-96 system. The dashed line is the standard deviation of the observation error.

It is also important to note that the higher rmse of the PF compared to the SEnKF and EnRDA is due to the problem of filter degeneracy which is further exacerbated by the presence of systematic errors in model forecasts (Poterjoy and Anderson,

2016). To alleviate this problem, one may investigate the use of methodologies suggested in recent years including the auxiliary particle filter where the weights of the particles at each assimilation cycle are defined based on the likelihood function from the next cycle using a pre-model run (Pitt and Shephard, 1999), the backtracking particle filter in which the analysis state is backtracked to identify the time step when the filter became degenerate (Spiller et al., 2008) as well as sampling from a transition density to pull back particles towards observations (van Leeuwen, 2010).

**4.2   Quasi-Geostrophic Model**

The multilayered quasi-geostrophic (QG, Pedlosky et al., 1987) model is known as one of the simplest circulation models capable of providing a reasonable representation of the mesoscale variability in geophysical flows. In its simplified form, the QG model describes the conservation of potential vorticity $\{\zeta_k\}_{k=1}^{K}$ in $K$ vertically-mixed vertical layers:

$$\left(\frac{\partial}{\partial t} + u_k \frac{\partial}{\partial \lambda} + v_k \frac{\partial}{\partial \phi}\right)\zeta_k = 0, \qquad k = 1, \dots, K,\tag{9}$$

where $u_k = -\dfrac{\partial \Psi_k}{\partial \phi}$ and $v_k = \dfrac{\partial \Psi_k}{\partial \lambda}$ represent the zonal and meridional components of the velocity field, obtained from the geostrophic approximation; $\{\Psi_k\}_{k=1}^{K}$ is the streamfunction in $K$ layers; and $\lambda$ and $\phi$ are the zonal and meridional coordinates, respectively.





For a two-layer QG model ($K = 2$), the potential vorticity at any time step is the sum of the relative vorticity, the planetary vorticity and the stretching term, given by:

$$\zeta_k = \nabla^2 \Psi_k + f + (1 - 2\delta_{2k})\frac{f_0^2}{g' h_k}(\Psi_2 - \Psi_1) \qquad k = 1, \ldots, 2, \tag{10}$$

where $\nabla^2(\cdot) = \dfrac{\partial^2(\cdot)}{\partial \lambda^2} + \dfrac{\partial^2(\cdot)}{\partial \phi^2}$ is the Laplace operator, $f = f_0 + \beta(\phi - \phi_0)$ is the Coriolis parameter linearly varying with the meridional coordinate $\phi$ ($\beta$-plane approximation), $f_0$ is the Coriolis parameter at mid-basin where $\phi = \phi_0$, $g' = \dfrac{g(\rho_2 - \rho_1)}{\rho_2}$ is the reduced value of the gravitational acceleration $g$, $\rho_k$ and $h_k$ are the density and thickness of the $k^{\text{th}}$ layer, respectively. The QG model has been the subject of numerous experiments to test the performance of DA techniques (Evensen, 1994b; Evensen and Van Leeuwen, 1996; Fisher and Gürol, 2017; Penny et al., 2019; Cotter et al., 2020).

### 4.2.1 Experimental Setup, Results and Discussion

Due to the high-dimensionality of the QG model and the well-known problem of filter degeneracy in the PF, we chose to omit its application on the QG model. Similar to the study conducted in (Evensen, 1992, 1994b), the streamfunction is chosen as the state variable for the DA experiments. The streamfunction field, at each vertical layer, is discretized over a uniform grid of dimension $m_\lambda \times m_\phi$ with spacing of $\Delta\lambda = \Delta\phi = 100$ km, where $m_\lambda = 65$ and $m_\phi = 33$. The model domain is assumed to have periodic boundaries along the zonal direction and free-slip conditions, that is, $v_k = 0, \forall k$, holds on the northern and southern boundaries. The standard model parameter values of $f_0 = 7.28 \times 10^{-5}$ s$^{-1}$, $\beta = 2 \times 10^{-11}$ m$^{-1}$ s$^{-1}$, and $g = 9.81$ m s$^{-2}$ are used. The total depth of the atmospheric column is set to 10 km with depths and densities of top and bottom layer as $h_1 = h_2 = 5$ km, and $\rho_1 = 1$ and $\rho_2 = 1.05$ kg m$^{-3}$, respectively. We first initialize the streamfunction in the two layers as a function of the zonal and meridional coordinates by setting $\Psi_1(\lambda, \phi) = -12.5 \times 10^6 \tan^{-1}\left(20(\phi/\Delta\phi - m_\phi/2)m_\phi^{-1}\right) - 1.25 \times 10^6 \sin\left(2\pi(\lambda/\Delta\lambda - 1)m_\lambda^{-1}\right)\sin^2\left(2\pi(\phi/\Delta\phi - 1)(m_\phi - 1)^{-1}\right)$ m$^2$ s$^{-1}$ and $\Psi_2(\lambda, \phi) = 0.3\,\Psi_1(\lambda, \phi)$.

From the initial value of the streamfunction field in each layer, potential vorticity is obtained using a nine-point second-order finite difference scheme to compute the Laplacian in Equation 10. The model in Equation 9 is then integrated with a time step of $\Delta t = 0.5$ hr using the fourth-order Runge-Kutta approximation to advect and obtain potential vorticity at internal grid points for the next time step. The streamfunction at the next time step is then calculated from this potential vorticity by solving the set of the Helmholtz equations (Equation 10). To avoid any form of initial transient behavior and to create vortex structures in the streamfunction, the QG model is integrated first for 720 time steps and then the endpoint of the run is used as the initial condition for subsequent DA experimentation.

The ground truth of the streamfunction is obtained by integrating the QG model with a time step of $\Delta t$ over a time period of $T = 0 - 15$ day in the absence of any model error. Observations are assumed to be available at an assimilation time interval of $24\Delta t$ or 12 hr. To construct observations, representative, random and systematic errors are applied to the ground truth. The representative error is applied by lowering the resolution of the ground truth through box averaging over a window of size $n_\lambda \times n_\phi$, where $n_\lambda = 5$ and $n_\phi = 3$. Then a heteroscedastic biased Gaussian noise with mean (standard deviation) $0.6 \times 10^6$ m$^2$ s$^{-1}$, equivalent to 33 (10%) of the mean magnitude of the ground truth is applied.

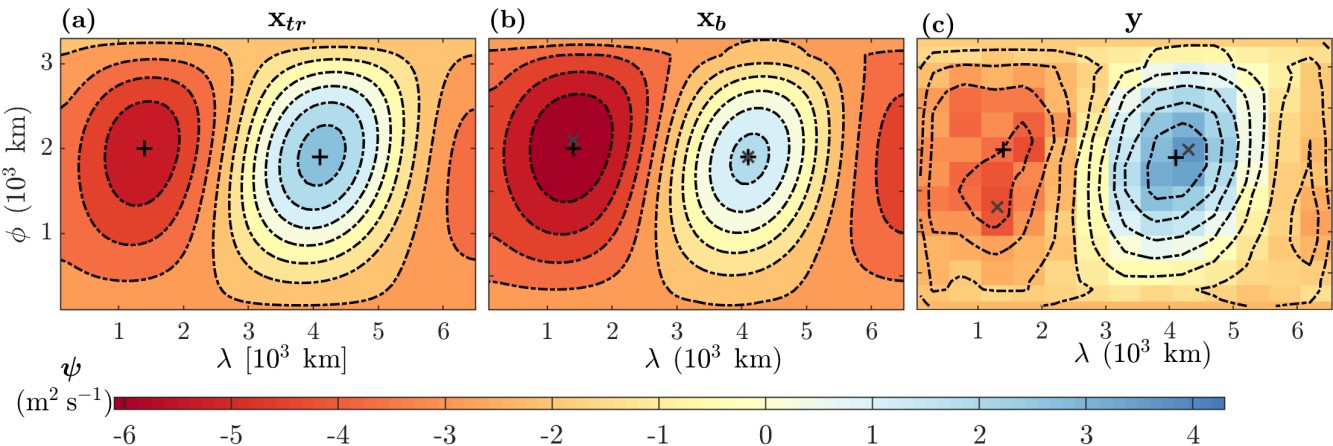

**Figure 6.** (a) The true state $\mathbf{x}_{tr}$, (b) background state $\mathbf{x}_b$, and (c) observations $\mathbf{y}$ for bottom layer field of streamfunction in the quasi-geostrophic model at first assimilation cycle $T = 12$ hr. The black plus (grey cross) signs show the location of the global extrema for the true state (background and observation).

To characterize the distribution of the background state, 50 ensemble members for both SEnKF and EnRDA are generated using model errors $\boldsymbol{\omega}_t \sim \mathcal{N}(0, \alpha \sigma_t^2 \, \mathbf{I}_{m_\lambda \times m_\phi})$ for each layer with $\sigma_0^2 = 10^8 \, \mathrm{m^4 \, s^{-2}}$ and $\sigma_t^2 = 5 \times 10^6 \, \mathrm{m^4 \, s^{-2}}$ for $t > 0$, where the factor $\alpha \in [0, 1]$ grows linearly from 0 at the northern and southern boundaries to 1 at mid-basin. To introduce systematic errors in the forecast, we utilize a multiplicative error of 0.015% in the QG model by multiplying the potential vorticity obtained from Equation 10 at every $\Delta t$ with a factor of 1.00015. At each assimilation cycle, $N = 500$ samples of the observations are obtained by perturbing the observations with the heteroscedastic Gaussian noise with standard deviation 10% of the mean magnitude of the ground truth.

In the SEnKF, to alleviate the well-known problem of undersampling (Anderson, 2012) and improve its performance, we utilize covariance inflation (Anderson and Anderson, 1999) and localization (Houtekamer and Mitchell, 2001; Hamill, 2001) as discussed in Appendix A2. For EnRDA, similar to the Lorenz-96 setup (Section 4.1.1), the displacement parameter is set to $\eta = 0.4$ through a cross-validation study based on a minimum rmse criterion as shown in Table 1. To increase the robustness of the inference about the results, the quality metrics are averaged using 10 simulations with different random realizations.

The true state, background state, and the observations of the bottom layer streamfunction at the first assimilation cycle $T = 12$ hr are shown in Figure 6. It can be seen that both the background state and the observations show possible systematic biases as the position and the values of their global extrema are significantly different from the ground truth.

The results of the DA experiments using the SEnKF and EnRDA at the first assimilation cycle for the bottom layer are also shown in Figure 7. It can be seen that, in the SEnKF, the streamfunction values are slightly overestimated, signaling the persistence of bias in the analysis state (Figure 7a). This is further evident as the analysis error field is coherent and structured (Figure 7b). On the other hand, it appears that EnRDA (Figure 7d) results in a more incoherent error field with a reduced bias (Figure 7e). The rmse for the EnRDA ($0.28 \times 10^6 \, \mathrm{m^2 \, s^{-1}}$) is lower than the one by the SEnKF ($0.46 \times 10^6 \, \mathrm{m^2 \, s^{-1}}$). However,

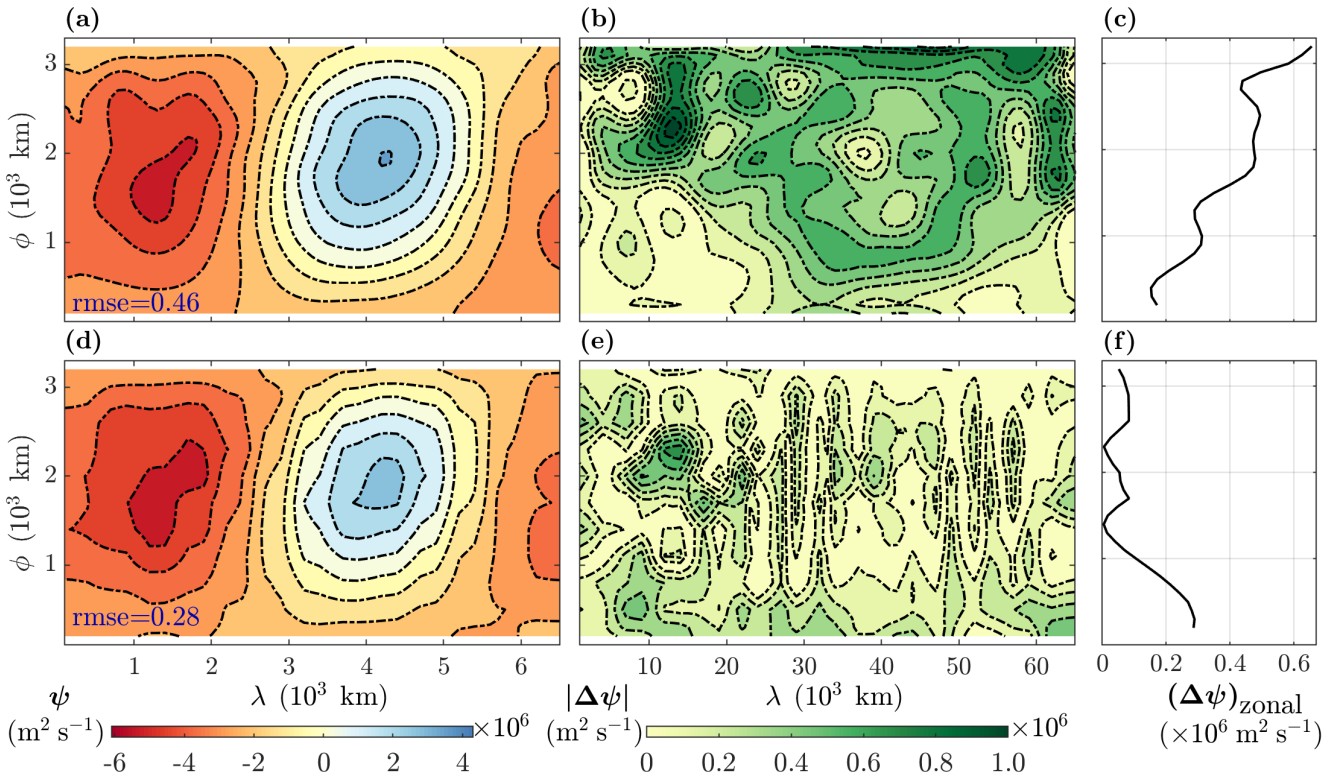

**Figure 7.** The streamfunction analysis state $\mathbf{x}_a$ by (a) Stochastic Ensemble Kalman Filter (SEnKF), and (d) Ensemble Riemannian Data Assimilation (EnRDA) as well as (b, e) their respective absolute error fields and (c, f) zonal mean of the error for the bottom layer of quasi-geostrophic model, at the first assimilation cycle $T = 12$ hr. The root mean squared error (rmse) values ($\times 10^6$ m$^2$ s$^{-1}$) for the entire fields are also reported in (a) and (d).

**Table 1.** Average root mean squared error (rmse) values as a function of the displacement parameter $\eta \in [0.25, 0.6]$ for Ensemble Riemannian Data Assimilation (EnRDA) from 10 independent simulations of the two-layer quasi-geostrophic model.

|  | rmse ($\times 10^6$ m$^2$ s$^{-1}$) | | | | | | | |
|---|---|---|---|---|---|---|---|---|
| $\eta$ | 0.25 | 0.30 | 0.35 | 0.40 | 0.45 | 0.50 | 0.55 | 0.60 |
| Top layer | 0.283 | 0.260 | 0.255 | 0.242 | 0.250 | 0.258 | 0.309 | 0.369 |
| Bottom layer | 0.211 | 0.198 | 0.194 | 0.189 | 0.206 | 0.222 | 0.294 | 0.368 |
| Average | 0.247 | 0.229 | 0.224 | 0.215 | 0.228 | 0.240 | 0.301 | 0.369 |

the difference between the two methods shrinks over $T = 0 - 15$ days and the mean analysis rmse over both layers by the EnRDA (SEnKF) reaches $0.21 \times 10^6$ ($0.25 \times 10^6$) m$^2$ s$^{-1}$. Furthermore, in the SEnKF, due to the presence of systematic error, the zonal mean of the absolute error is consistently higher than that of the EnRDA, see (Figure 7c and f).

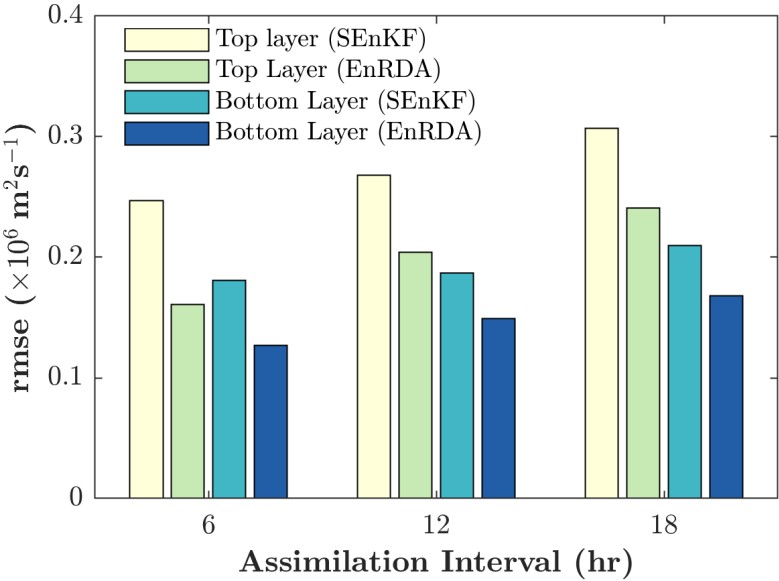

**Figure 8.** The average root mean squared error (rmse) values as a function of assimilation intervals 6, 12 and 18 hr in the Stochastic Ensemble Kalman Filter (SEnKF) and Ensemble Riemannian Data Assimilation (EnRDA) for the two-layer quasi-geostrophic model.

We further examined the performance of the EnRDA and the SEnKF on the QG model with a $\pm 50\%$ change in the assimilation interval of 12 hr as shown in Figure. 8. To make the comparison fair between different assimilation intervals which have a different number of assimilation cycles and to eliminate the impact of transient behavior, we only report the statistics for the last 15 assimilation steps. With the increase in assimilation interval, the systematic error grows in the forecast largely due to the multiplicative error being added to the forecast at every time step. Therefore, as is expected, with the increase in assimilation interval, the rmse grows monotonically and the performance of the DA methodologies degrades. However, the EnRDA demonstrates consistent improvement over a bias-blind implementation of the SEnKF (20–33%) across the range of assimilation intervals. On average, using cluster with 24 cores and a clock rate of 2.5 GHz, it took around 3.5 hr to complete one independent simulation on Q-G model for EnRDA compared to 2.5 hr for EnKF each with 50 ensemble members.

## 5 Summary and Concluding Remarks

In this study, we demonstrated that data assimilation (DA) over the Wasserstein space through the Ensemble Riemannian DA (EnRDA, Tamang et al. (2021)) can be properly scaled and result in improved predictability of non-Gaussian geophysical dynamics at relatively high dimensions, under systematic errors. In particular, we applied the EnRDA to the 40-dimensional chaotic Lorenz-96 system and a two-layer quasi-geostrophic representation of atmospheric circulation and compared its results with the Stochastic Ensemble Kalman Filter and the Particle Filter with comparable ensemble size. Under the made assumptions and experimental settings, EnRDA improved the root mean squared error by almost 20%(30%) for the Lorenz-96 (Q-G)





model, when compared to the classic Euclidean DA techniques. We need to emphasize that in the absence of systematic errors,
Euclidean DA methodologies definitely demonstrate improved performance over EnRDA in terms of the root mean squared
error. Despite the reported improvements, further comprehensive comparisons with bias-aware versions of the Euclidean DA
methodologies are required to fully characterize the pros and cons of DA over the Wasserstein space.

One of the major weaknesses of the presented methodology in its current form is that all dimensions of the problem are
assumed to be observable. This is an important issue when it comes to the assimilation of sparse data. Future research is
needed to address partial observability in DA over the Wasserstein space. A possible direction is through multi-marginal
optimal mass transport (Pass, 2015), which could enable to couple different dimensions of the problem and propagate the
information content of sparse observations to unobserved dimensions. Moreover, currently, the displacement parameter is
constant across multiple dimensions of the problem. Future research is needed to understand how the displacement parameter
can be estimated differently depending on the error structure across different dimensions of the state space. Another option is to
perform the EnRDA only in that part of the state space that is directly observed and use the ensemble covariance to update the
unobserved part of state space, similar to a SEnKF. We anticipate that expanding the application of the presented methodology
for assimilating satellite data into land-atmosphere models could be another promising future direction of research given the
fact that these models are often markedly biased (Dee and Da Silva, 1998; Chepurin et al., 2005; De Lannoy et al., 2007; Lin
et al., 2017).

**Appendix A: Appendix**

**A1   Sinkhorn's Algorithm for Optimal Mass Transport**

To solve the regularized optimal mass transport problem in Equation 7, we utilize Sinkhorn's algorithm (Sinkhorn, 1967). To
that end, first, the Lagrangian form of the Equation 7 using two Lagrange multipliers $\mathbf{a} \in \mathbb{R}^M$ and $\mathbf{b} \in \mathbb{R}^N$ is obtained as
follows:

$$\mathcal{L}(\mathbf{U}, \mathbf{a}, \mathbf{b}) = \text{tr}(\mathbf{C}^{\text{T}}\mathbf{U}) - \gamma \, \text{tr}\big(\mathbf{U}^{\text{T}}[\log(\mathbf{U} - \mathbb{1}_M \mathbb{1}_N^{\text{T}})]\big) - \mathbf{a}^{\text{T}}(\mathbf{U}\mathbb{1}_N - \mathbf{p}_x) - \mathbf{b}^{\text{T}}(\mathbf{U}^{\text{T}}\mathbb{1}_M - \widetilde{\mathbf{p}}_{y|x}). \tag{A1}$$

Now, we set the first-order derivative of the Lagrangian form in Equation A1, with respect to $(i,j)^{\text{th}}$ element of the joint
distribution $(u_{ij})$ to zero:

$$\frac{\partial \mathcal{L}(\mathbf{U}, \mathbf{a}, \mathbf{b})}{\partial u_{ij}} = c_{ij} + \gamma \log(u_{ij}) - a_i - b_j = 0 \qquad \forall i, j, \tag{A2}$$

which ultimately leads to $u_{ij} = \exp\left(\dfrac{a_i}{\gamma}\right) \exp\left(-\dfrac{c_{ij}}{\gamma}\right) \exp\left(\dfrac{b_j}{\gamma}\right)$. This can be rewritten in a matrix form as $\mathbf{U}^a =$
$\text{diag}(\mathbf{s}) \mathbf{V} \text{diag}(\mathbf{t})$, where $\left\{\mathbf{V} \in \mathbb{R}_+^{M \times N} : v_{ij} = \exp\left(-\dfrac{c_{ij}}{\gamma}\right)\right\}$ is the Gibb's kernel of the cost matrix $\mathbf{C}$, and $\mathbf{s} \in \mathbb{R}^M, \mathbf{t} \in \mathbb{R}^N$





are the unknown scaling vectors. The notation $\mathrm{diag}(\mathbf{x}) \in \mathbb{R}^{M \times M}$ represents a diagonal matrix with its diagonal entries provided by $\mathbf{x} \in \mathbb{R}^M$.

By setting the derivatives of the Lagrangian with respect to the Lagrange multipliers as zero we recover the two conditions, which we can write as $\mathbf{p}_x = \mathrm{diag}(\mathbf{s})\mathbf{V}\,\mathrm{diag}(\mathbf{t})\mathbb{1}_N$ and $\widetilde{\mathbf{p}}_{y|x} = \mathrm{diag}(\mathbf{t})\mathbf{V}^{\mathrm{T}}\,\mathrm{diag}(\mathbf{s})\mathbb{1}_M$ leading to:

$$\mathbf{s} = \mathbf{p}_x \oslash (\mathbf{V}\,\mathbf{t}) \qquad \text{and} \qquad \mathbf{t} = \widetilde{\mathbf{p}}_{y|x} \oslash (\mathbf{V}^{\mathrm{T}}\,\mathbf{s}), \tag{A3}$$


where the notation $\mathbf{x} \oslash \mathbf{y}$ represents a Hadamard element-wise division of equal length vectors. The form presented in Equation A3 is known as the matrix scaling problem (Borobia and Cantó, 1998) and can be efficiently solved iteratively:

$$\mathbf{s}^{(i)} = \mathbf{p}_x \oslash (\mathbf{V}\,\mathbf{t}^{(i-1)}) \qquad \text{and} \qquad \mathbf{t}^{(i)} = \widetilde{\mathbf{p}}_{y|x} \oslash (\mathbf{V}^{\mathrm{T}}\,\mathbf{s}^{(i)}), \tag{A4}$$

where $i$ is the iteration count and the algorithm is initialized with a positive vector $\mathbf{t}^{(0)} = \mathbb{1}_N$. In our implementation, we set

the iteration termination criterion as $\dfrac{\left\| \mathbf{s}^{(i)} - \mathbf{s}^{(i-1)} \right\|_2}{\left\| \mathbf{s}^{(i-1)} \right\|_2} \le 10^{-4}$ or $i > 300$. After the convergence of the solution for $\mathbf{s}$ and $\mathbf{t}$, the optimal joint distribution can be obtained as $\mathbf{U}^a = \mathrm{diag}(\mathbf{s})\mathbf{V}\,\mathrm{diag}(\mathbf{t})$.

## A2  Covariance Inflation and Localization in Ensemble Kalman Filter

The ensemble size in the Stochastic Ensemble Kalman filter (SEnKF), if much smaller than the state dimension, such as in the presented case of the quasi-geostrophic model, leads to underestimation of the forecast error covariance matrix and

subsequently filter divergence problems. To alleviate this problem, a covariance inflation procedure can be implemented by multiplying the forecast error covariance matrix by an inflation factor $\tau > 1$ (Anderson and Anderson, 1999) where its optimal value depend on the ensemble size (Hamill et al., 2001) and other characteristics of the problem at hand.

The covariance localization procedure in the SEnKF further attempts to improve its performance by ignoring the spurious long-range dependence in the ensemble background covariance by applying a prespecified cutoff threshold on the correlation

structure of the field. An SEnKF equipped with a tuned localization procedure can be efficiently used in high-dimensional atmospheric and ocean models even with less than 100 ensemble members (Anderson, 2012). The covariance localization in an SEnKF is accomplished by modifying the Kalman gain matrix $\mathbf{K} \in \mathbb{R}^{m \times m}$ through implementation of a Hadamard element-wise product of the forecast error covariance matrix $\mathbf{B} \in \mathbb{R}^{m \times m}$ with a distance-based correlation matrix $\boldsymbol{\rho} \in \mathbb{R}^{m \times m}$:

$$\mathbf{K} = (\boldsymbol{\rho} \odot \mathbf{B})\mathbf{H}^{\mathrm{T}} \big( \mathbf{H}(\boldsymbol{\rho} \odot \mathbf{B})\mathbf{H}^{\mathrm{T}} + \mathbf{R} \big)^{-1}, \tag{A5}$$

where $\mathbf{X} \odot \mathbf{Y}$ represent the Hadamard element-wise product between equal size matrices $\mathbf{X}$ and $\mathbf{Y}$.





Following the work of Gaspari and Cohn (1999), we utilized the fifth-order piece-wise rational function that depends on a single length scale parameter $d$ and an Euclidean distance matrix $\{\mathbf{L} \in \mathbb{R}^{m \times m} : l_{ij} = \|x_i - x_j\|_2\}$ for obtaining the $(i,j)^{\text{th}}$-element of the localizing correlation matrix $\boldsymbol{\rho}$:

$$
\rho_{ij} = \begin{cases}
-\dfrac{1}{4}r^5 + \dfrac{1}{2}r^4 + \dfrac{5}{8}r^3 - \dfrac{5}{3}r^2 + 1, & 0 \leq r \leq 1, \\[2em]
\dfrac{1}{12}r^5 - \dfrac{1}{2}r^4 + \dfrac{5}{8}r^3 + \dfrac{5}{3}r^2 - 5r + 4 - \dfrac{2}{3}r^{-1}, & 1 < r \leq 2, \\[2em]
0, & r > 2,
\end{cases}
\tag{A6}
$$

where $r = \dfrac{l_{ij}}{d}$, and $d$ is the length scale.

In our implementation of the SEnKF in the QG model, the inflation factor and length scale were chosen between $\tau = 1.01 - 1.08$ and $d = 400 - 1800$ [km] respectively depending on the experimental setup through trial and error analysis to minimize the root mean squared error.





## Acknowledgements

Data archiving is underway at the Data Repository for University of Minnesota (https://conservancy.umn.edu/handle/11299/166578). The first and second author acknowledge the support by grants from the National Aeronautics and Space Administration (NASA) Remote Sensing Theory program (RST, 80NSSC20K1717), the Interdisciplinary Research in Earth Science program (IDS, 80NSSC20K1294) and the New (Early Career) Investigator Program (NIP, 80NSSC18K0742). The third author acknowledges support from the European Research Council for funding via the Horizon2020 CUNDA project under number

694509. The fourth author also acknowledges support from National Science Foundation (NSF, DMS1830418). The authors also acknowledge the Minnesota Supercomputing Institute (MSI) at the University of Minnesota for providing resources that contributed to the research results reported within this paper.



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
