# Peer review of "Ensemble Riemannian Data Assimilation: Towards High-dimensional Implementation"

_Nonlinear Processes in Geophysics, 2021_

## Author Comment (AC1)

**Ensemble Riemannian Data Assimilation: Towards High-dimensional Implementation**

By: Sagar K. Tamang, Ardeshir Ebtehaj, Peter J. van Leeuwen, Gilad Lerman, and Efi Foufoula-Georgiou

**Responses to review comments (Reviewer 1)**

**General comments**:
The authors extend their previous research from low-dimensional to a relatively higher dimensional problem. They include Lorenz-96 model and a two-layer quasi-geostrophic model as examples of medium-dimensional problem. The analysis is inferred from (optimized) joint distribution that couples possibly non-Gaussian probability distributions (PD) of the background PD with observation PD, which, according to authors, enables formal reduction of systematic biases. The system is compared to particle filter and stochastic Kalman filter, with results suggesting the mean squared analysis error can be reduced by 20%-30%. The manuscript is well written, and the figures are adequate.

Reply: We very much appreciate the reviewer for providing us with constructive feedback. The feedback helped us to improve the manuscript. In the revised manuscript with the track-change on, the blue colored text is the updated text in response to the reviewer's feedback. Please also find item-by-item replies to the comments as follows.

Comments:

1. Title: Including "high-dimensional" in title is presumptuous in my view. There is a long way between the dimensions considered here and in the realistic NWP system. The actual state vector dimensions are 40 for Lorenz-96 and 4,290 for QG model, while a realistic NWP model deals with a state dimension of the order of $10^8$. This is at least 5 orders of magnitude larger than QG model. Technically they may be correct by saying "towards highdimensional" but going from Lorenz-63 with state dimension 3 to Lorenz-96 with state dimension 40 is hardly an important step towards realistic state dimension 100,000,000. I suggest using words "moderate" or "intermediate" instead of "high-dimensional", or completely changing the title to reflect better the experiments conducted in the manuscript.

   Thank you for this comment. We agree that the Numerical Weather Prediction models often deal with state dimension in order of hundreds of millions. However, the goal of this paper is to rather provide a proof of concept on the applicability of the EnRDA to the high-dimensional problems. It is well known that fully nonlinear data assimilation (DA) methodologies, such as the widely used particle filter (PF), require the number of particles to grow exponentially with the state dimension (Snyder et al., 2008). For example, the necessary ensemble size for a PF increases exponentially from $10^6$ to $10^{11}$ when state dimension doubles from 100 to 200 (Snyder et al., 2008). However, this is not the case with the EnRDA. By applying the EnRDA methodology to 40-dimensional Lorenz-96 and 4290-dimensional Quasi-geostrophic model both with just 50 ensemble members each, we aimed to demonstrate that the EnRDA methodology has the potential to be used in high-dimensional Earth system models without

significant computational burden. For example, in numerous articles (e.g., Poterjoy, 2016; Penny and Miyoshi, 2016), the authors used the word "high-dimensional", while testing the proposed DA methodologies on the 40-dimensional Lorenz-96 dynamical system. Having said that, in response we changed the title to " Ensemble Riemannian Data Assimilation: Towards Large-Scale Dynamical Systems". We are open to any suggestion from the reviewer as well. At the same time, we have updated the text to made this clear in the introduction that there is still a long way to make this approach available for large-scale Earth system models. Please see lines 70–72.

2. Multivariate aspect: All problems considered in this (and previous) manuscript are univariate. This clearly reduces the complexity of the problem and does not address the possibility of extending this methodology to more realistic situation. Unless the authors want to include multivariate experiments and results, they should clearly state these systems are univariate and elaborate on potential difficulties of applying the system to multivariate problems.

Thank you for this comment. A possible solution to this problem can be obtained by rather utilizing the Mahalanobis or the weighted Euclidean distance (Olver et al., 2006, p.133) in lieu of the Euclidean distance in the ground transportation cost matrix $\{\widetilde{\mathbf{C}} \in \mathbb{R}^{M \times N} : \tilde{c}_{ij} = ||\mathbf{x}_i - \mathbf{y}_j||^2_{\mathbf{T}^{-1}}\}$, where $\mathbf{T} \in \mathbb{R}^{M \times N}$ represents a covariance matrix that allows cross-dimensional penalization of the mismatch between multivariate representation of the state space. Future research is needed to test such an extension of the ground cost matrix on DA problems. Please see lines 389–393, where we have updated the text in response to this feedback.

3. Non-Gaussian probability distribution of errors: Although authors imply throughout the manuscript that the presented methodology is suitable for non-Gaussian errors, they use only Gaussian probability distribution in the Lorenz-96 and QG-model experiments (e.g., p.14, L.324-326; p.5, L.142; p.16, L.354-355, etc). I believe that including experiments with skewed non-Gaussian probability distributions, such as Lognormal, Gamma, or Beta, would strengthen the authors' arguments and improve the presentation of the new method. Skewed distributions would automatically address the systematic bias and non-Gaussian errors. If the authors would like to postpone such experiments for the future, that is okay, but they need to clearly state this as a limitation of the current experimental setup.

We appreciate the comment. We would like to clarify that by non-Gaussianity throughout the manuscript, we were referring to the non-Gaussian distribution of the prior, posterior and observations, and not necessarily the associated errors. Please see lines 68–70. It is known that any nonlinear dynamics, such as Lorenz-63, would result in a non-Gaussian distribution of the state regardless of the error distribution (Bocquet et al., 2010). However, we agree that an additional implementation of the En-RDA under the presence of non-Gaussian observation errors will enrich the presentation. Therefore, we have conducted additional DA experimentation using an additive Laplace-distributed observation error similar to the study conducted by Lei and Bickel (2011); Spantini et al. (2019). Please note that using non-symmetric distributions

(e.g., gamma), for representation of errors, renders the classic DA such as EnKF sub-optimal and does not let us to have a level field for comparing EnRDA with classic methods, which are minimum mean squared estimator under Gaussian error. Please see lines 285–289 and 352–355 (new version) where we have added the details of the experimentation (also copied below for convenience).

To further test the efficiency of EnRDA, another configuration of the Lorenz-96 is implemented using a Laplace-distributed observation error at each assimilation interval of $10\Delta t$ with variance $\sigma_{obs}^2 = 2$ (Lei and Bickel, 2011; Spantini et al., 2019). Similar to the setting implemented earlier in the case of Gaussian observation error, 50 (5000) ensemble members (particles) for the SEnKF and EnRDA (PF) are generated. On average, the EnRDA reduces the rmse by 26% (47%) compared to the SEnKF (PF) using 50 random realizations.

4. I feel that possible benefits of bias correction in SEnKF have not been sufficiently explored to make a more realistic comparison between the experiments. While the authors acknowledge this, I think implementing some basic bias correction methodology (e.g., moving average etc) in SEnKF would make the strength of EnRDA method clearer.

Thank you. We agree that adding a bias correction to SEnKF or PF will enrich the paper. However, currently the message is that unlike the Euclidean distance, the Wasserstein metric is geodesic and can naturally account for the mismatch between all moments of two probability distributions and thus can correct for systematic biases without any *ad hoc* bias correction methodologies. Future research can be devoted to comparing bias corrected SEnKF/PF with the DA over Wasserstein space. Please see lines 374–376, where we addressed this comment.

5. p.1, L. 11: Although in NWP practice DA is mostly about optimizing the initial conditions, I do not agree that DA "science" is only about initial conditions: it is about estimating the probability Density Function (PDF) or its discrete equivalent (e.g., probability mass function). I am not requesting this, only suggesting.

We appreciate the feedback. Looking at the geophysical forecast models as an initial value problem, the goal of DA (Kalnay, 2003, p.12) is to reduce uncertainty of geophysical forecasts by improving the initial condition of the geophysical models. Certainly we agree with the reviewer that in a much broader sense, DA falls into a larger group of class of methodologies for optimal estimation of the probability distribution of the state given observations and previous time model forecasts. Please see the lines 10–12 where we have updated the text to address this.

6. p.1, L.14-15: Not sure that Is DA formulation is a "penalization of second-order statistics"? It is a penalization of the cost function, which defines weighted (Euclidian) distances, as authors mention. However not sure why/how a penalization of (Euclidian) distances is a "penalization of second-order statistics" as it could be implied from text. Please explain.

Thank you for this comment. The text should read "penalization based on second-order statistics of error". The cost function is comprised of the weighted Euclidean

distances with respective inverse of the error covariance matrices (second-order error statistics). Indeed, if we assume that the error covariance matrices are diagonal, we are minimizing sum of squared error or variance of the analysis error. Please see the updated text in lines 12–16.

7. p.9, L.220: the model error term omega-t has zero bias (p.8, L.213). Why is there a systematic error in prediction? Please explain.

The systematic error in our DA experiments for Lorenz-96 is introduced through an incorrect model forcing of $F_m = 6$ (instead of the true model forcing $F = 8$) as noted in lines 215–216 (old version) and 220–221 (revised version).

8. p.13, L. 318; p.14, L.325: "heteroscedastic" implies that variance (second-order feature) depends on the point. However, systematic bias is a first order feature. Please explain.

We agree with the reviewer's comment that the heteroscedasticity and bias are different-order features. By heteroscedastic biased Gaussian noise (line 318 and 325 (old version)), we meant a Gaussian noise which has a pre-specified bias and a variance that changes depending on the magnitude of the state variable at that particular instance of time. We have modified the text to improve readability, please see the revised text in lines 328–329.

We would like to take this opportunity and once gain thank you for taking the time and providing us a thorough review of the manuscript. We did our best to incorporate your comments into the new revision. We hope that the replies and changes we made in the manuscript meet your expectations.

**Responses to review comments (Reviewer 2)**

The authors extend their recent work on Ensemble Riemannian Data Assimilation (EnRDA) to high-dimensional systems. Specifically, "high" here refers to a dimension size where optimal mass transport (OMT) would become computationally intractable using standard approaches with OMT to approximate the resulting probability distributions. The numerical experiments considered have DA test problems with dimensions of 40 and 4290, and each uses 50 ensemble members for the EnRDA implementation. The EnRDA results in the numerical experiments demonstrate an improvement over a stochastic EnKF when both are applied to DA test problems that are constructed to have strong systematic bias.

This paper is clear and well written. In particular I commend the authors for their nice introduction to OMT which will be accessible to readers familiar with DA, but not with OMT.

Reply: We very much appreciate positive review feedback and comments on the manuscript. We have included a track-change color coded version in which red colored text is the updated text in response to the reviewer's feedback. Please also find item-by-item replies to your comments as follows.

I think the word "towards" in the title demands a more extensive "Summary and Concluding Remarks" section. Clearly the issue of OMT's need to observe all dimensions is a serious challenge for EnRDA to be applied to assimilate real data into high-dimensional models. Although you do discuss this, the discussion is rather brief. Further, for systems of dimensions $10^6 - 10^8$, it seems that for EnRDA to be applied, some kind of dimension reduction would have to be introduced. How might EnRDA and dimension reduction work together? What are the challenges? A paragraph or two thinking about what might be tried on this front would be useful.

Thank you for this comment. One of the major advantages of the EnRDA is that unlike the fully-nonlinear data assimilation (DA) methodologies, such as the particle filter (PF), the computational complexity does not scale exponentially with the dimension of the problem. We utilized the same ensemble size of 50 for both 40-dimensional Lorenz-96 and 4290-dimensional Quasi-geostrophic model and consistently performed better than both PF and stochastic Ensemble Kalman filter. We aimed to present this as a proof of concept and certainly further research is needed to expand the approach to large-scale Earth system models. We also share your view that conducting a dimensionality reduction prior to EnRDA is a great idea and needs to be addressed in future research efforts. Please see the added paragraph in lines 394–399 in Conclusion Section.

Minor comments:

1. Would M and N typically be different in practice? For presentation, it totally makes sense to keep them distinct. Yet if M=N in practice for EnRDA, you might want to mention this explicitly to the reader.

   Thank you for this comment. The EnRDA methodology allows $M$ and $N$ to be different, however, one can also choose them equal based on the problem at hand. The optimal transportation plan matrix $\mathbf{U}$ can therefore, be a rectangular matrix. However, since the computational cost of the EnRDA methodology is tied to the size of $\mathbf{U}$, $M$ and $N$ needs to be chosen to adjust the trade-off between accuracy and computational cost. Please see lines 167–169 where we have responded to this feedback in the text.

2. There seems to be a word missing on line 255, "..enables to obtain".

   Thank you for this comment. We have corrected the typo.

Once again, we would like to take this opportunity and thank the reviewer for providing us constructive feedback. The feedback helped us to improve the presentation and readership of the manuscript.

**References**

Bocquet, M., Pires, C. A., and Wu, L.: Beyond Gaussian statistical modeling in geophysical data assimilation, Monthly Weather Review, 138, 2997–3023, 2010.

Kalnay, E.: Atmospheric modeling, data assimilation and predictability, Cambridge university press, 2003.

Lei, J. and Bickel, P.: A moment matching ensemble filter for nonlinear non-Gaussian data assimilation, Monthly Weather Review, 139, 3964–3973, 2011.

Olver, P. J., Shakiban, C., and Shakiban, C.: Applied linear algebra, vol. 1, Springer, 2006.

Penny, S. G. and Miyoshi, T.: A local particle filter for high-dimensional geophysical systems, Nonlinear Processes in Geophysics, 23, 391–405, 2016.

Poterjoy, J.: A localized particle filter for high-dimensional nonlinear systems, Monthly Weather Review, 144, 59–76, 2016.

Snyder, C., Bengtsson, T., Bickel, P., and Anderson, J.: Obstacles to high-dimensional particle filtering, Monthly Weather Review, 136, 4629–4640, 2008.

Spantini, A., Baptista, R., and Marzouk, Y.: Coupling techniques for nonlinear ensemble filtering, arXiv preprint arXiv:1907.00389, 2019.